# Acute Foot Drop Caused by Intraneural Ganglion Cyst of the Peroneal Nerve: Literature Review and Case Report

**DOI:** 10.3390/jpm13071137

**Published:** 2023-07-14

**Authors:** Giuseppe della Vecchia, Alfonso Baldi, Maria Beatrice Passavanti, Angela Lucariello, Antonio De Luca, Paolo De Blasiis

**Affiliations:** 1Department of Women, Child, General and Specialistic Surgery, University of Campania “Luigi Vanvitelli”, 80138 Naples, Italy; dellavecchia-g@libero.it (G.d.V.); mariabeatrice.passavanti@unicampania.it (M.B.P.); 2Department of Environmental, Biological and Pharmaceutical Sciences and Technologies, University of Campania “L. Vanvitelli”, 81100 Caserta, Italy; alfonso.baldi@unicampania.it; 3Department of Sport Sciences and Wellness, University of Naples “Parthenope”, 80100 Naples, Italy; angela.lucariello@gmail.com; 4Department of Mental and Physical Health and Preventive Medicine, Section of Human Anatomy, University of Campania “Luigi Vanvitelli”, 80131 Naples, Italy; antonio.deluca@unicampania.it

**Keywords:** acute foot drop, ganglion cyst, peroneal nerve

## Abstract

Background: Foot drop (FD) is characterized by an inability to lift the foot against gravity because of dorsiflexor muscle weakness. The aim of the present study is to report a clinical case of acute non-traumatic FD in patients with peroneal intraneural ganglion, after performing a scoping review on the methodological management of this disease. Methods: We performed a review of the literature and reported the case of a 49-year-old man with acute FD caused by an intraneural ganglion cyst of the peroneal nerve. Results: Out of a total of 201 articles, 3 were suitable for our review beyond our case report. The acute FD caused by peroneal intraneural ganglion can be managed by a careful clinical–instrumental differential diagnosis. A targeted surgery with subsequent rehabilitation produced a satisfactory motor recovery. Conclusions: Acute FD requires an appropriate diagnostic–therapeutic framework to identify and effectively treat the causes in order to promote complete recovery

## 1. Introduction

Foot drop (FD) is a clinical impairment characterized by an inability to lift the foot against gravity because of a dorsiflexor muscle weakness [1]. Patients affected by foot drop show a typical abnormal gait pattern with compensatory hyperflexion of the hip and knee joints associated with internal rotation of the foot in the transverse plane [2], which could be responsible for further injuries or falls [3]. FD can be unilateral or bilateral in relation to the causes that may concern central or peripheral nervous systems or the dorsiflexor muscle directly [1], for example, patients with multiple sclerosis, cerebral palsy, cerebrovascular disease, plexopathy, L5 radiculopathy or sciatic neuropathy [4]. A further cause of FD is fibular neuropathy, sometimes due to the presence of an intraneural ganglion (IG) [5], which commonly involves the common fibular nerve [6] as well as, less frequently, the ulnar and radial nerves [7]. The common peroneal nerve originates at the level of the superior angle of the popliteal fossa, descends in the direction of the leg, skirting the head of the fibula, then perforates the long peroneus muscle and finally divides into its terminal branches, superficial and deep peroneal nerve, where intraneural ganglion cysts may occur [5]. The clinical presentation of an intraneural ganglion may include the presence of a palpable mass, pain in the area of the cyst, hypoesthesia and variable force deficits in the affected nerve territory. Frequently, the onset of FD can be secondary to an acute traumatic event that induces the onset of paralysis [8,9]. Nevertheless, in isolated cases, FD may be a primary symptom with an acute onset and in the absence of a previous traumatic event in a patient who is apparently healthy [10]. Therefore, a careful diagnostic therapeutic procedure must be performed, and multidisciplinary management for an adequate approach and treatment is necessary [1].

The aim of the present study is to report a clinical case of acute non-traumatic foot drop in patients with peroneal intraneural ganglion, showing our management and performing a scoping review on the diagnostic therapeutic procedures of this disease.

## 2. Materials and Methods

### 2.1. Research Protocol

A literature review was performed, including all articles referred to patients (with an age range of 0 to 99 years) affected by acute nontraumatic FD secondary to peroneal nerve intraneural ganglion cyst. Moreover, a clinical case concerning the diagnostic therapeutic management of our patients with same disease was also presented. The review was structured in accordance with the guidelines dictated by the “PRISMA statement 2020” [11] (reporting product for systematic reviews and meta-analyses) and CARE [12], used, respectively, for drafting the systematic review of the literature and for drafting the case report. The main objective of the study is to create an overview of the methodological management of acute non-traumatic foot drop in patients with peroneal intraneural ganglion with particular reference to diagnostic pitfalls on the basis of data in the literature and on our clinical experience. As a secondary objective, we tried to raise awareness regarding the possible causes of acute-onset foot drop.

### 2.2. Search Strategy, Eligibility Criteria and Data Extraction

The search phase for potentially eligible trials began on 10 December 2022 and ended on 18 December 2022, performed by two independent reviewers. In the review phase, both reviewers independently analyzed all the titles and abstracts obtained from the search strings used in the various databases. Three electronic databases were queried: Tripdatabase, Pubmed and Clinicaltrial.Gov. Once the PICO (Population, Intervention, Comparison, Outcomes) query was formulated, the strings were built using the (MESH) database and Boolean indicators by associating all the possibilities of interaction between the various synonyms of the keywords, such as “peroneal ganglion”, “ganglion cyst”, “peroneal intraneural ganglion” and “foot drop”. Articles that did not fit our research question, for example, describing only non-acute-onset cases, post-traumatic cases, articles not accessible or not in English, as well as incomplete and duplicate studies, were excluded.

## 3. Results

### 3.1. Study Selection

The sequence of steps used in the review is summarized by the QUORUM flowchart (Figure 1). A total of three articles met the inclusion criteria. Thus, 198 studies were excluded for the reasons detailed in Figure 1. All subjects included who had acute-onset foot drop secondary to an interneural ganglion cyst of the peroneal nerve subsequently underwent neurosurgical intervention using cystic decompression. A summary of the included studies is included in Table 1.

### 3.2. Case Report

#### 3.2.1. Evaluation

A 49-year-old man, a worker, was evaluated to our outpatient clinic of Second Neurology at the University of Campania “Luigi Vanvitelli”, in November 2022. He reported a severe weakness of the ankle-dorsiflexor muscle associated to paresthesia of the lateral region of the right leg, occurring on the morning with an acute onset, about ten days ago, while getting into a car. He also reported a remote pathological history of arterial hypertension treated by beta-blocker and ACE-inhibitor drugs, a previous episode of left lumbosciatica, no further significant previous pathologies, no family history of specific pathologies and no previous or recent traumatic events. The patient underwent magnetic resonance imaging of the lumbosacral spine, which showed “severe disc-osteophytic protrusions L3/L4, L4/5 and L5/S1 with diffuse arthritic thickening of the facet joints and yellow ligaments on the arthritic-interpophyseal base, with consequent stenosis of the canal and neuroforamina”. He was previously treated by a neurological colleague with NSAIDs, Pregabalina and corticosteroids for “acute right lumbar sciatica”. After 10 days of treatment without beneficial effects, the patient came to our outpatient clinic, where a careful clinical examination revealed paresthesia in the lateral region of the right leg with associated pain symptoms (Numerical Rating Scale (NRS) 8/10), right foot drop with significant strength deficit of the tibialis anterior (TA) muscle (MRC = 1/5), of Extensor Digitorum Common (EDC) and Extensor Hallucis Longus (EHL) muscles (MRC = 0/5) and of peroneal muscles (MRC = 2/5), while the strength of all other lower limb muscles was normal (MRC = 5/5). Moreover, the active Range of Motion (RoM) of the ankle was 25° of plantiflexion, with an inability for dorsiflexion (Figure 2a). Negative Lasegue and Wasserman tests, tendon reflex normal bilaterally, preserved trophism of the quadricep and bilateral triceps surae muscle were found; Tinel test was performed laterally on the head of the right fibula, resulting as positive. Bedsite ultrasound was then performed with a multifrequency linear probe on the external popliteal sciatic nerve, starting from the popliteal fossa and throughout its course, and showed an oblong, lobulated (42.7 × 16.73 mm) anechoic formation, with homogeneous non-corpuscular content and well-defined margins located near the right external popliteal sciatic nerve, visualized near the posterolateral region of the fibula head, not captured on color Doppler (Figure 3a,b). Magnetic resonance imaging of the knee confirmed the presence of an oblong cystic formation approximately 5 cm in craniocaudal diameter of probable neurogenic nature that cannot be dissociated from the nerve sheath with fluid distension of the tibiofibular recess (Figure 3c,d). The subsequent electromyographic examination highlighted signs of marked neurogenic suffering, very severe in the extensor digitorum brevis muscle, where no voluntary electrical activity is recorded.

#### 3.2.2. Treatment

Open excision was performed due to important motor and axonal deficits, acute onset of symptoms, low tendency for symptom regression, high recurrence rate with voiding and corticosteroid infiltration and no response to the rehabilitation treatment with electrostimulation and physiotherapy. Informed consent was obtained from the patient for this suggested treatment. A curved incision extending proximally and distally from the superior tibiofibular joint was, therefore, applied, sufficient for complete exposure of the peroneal nerves, which were identified and isolated (Figure 3e,f). After opening the epineurium, the ganglion cyst was then emptied, which showed a clear gelatinous content (Figure 3e,f); a cylindrical fragment of about 4 cm was then dissected and removed and then sent for anatomical–pathological examination, which will confirm the presence of a mucoid cyst (so-called ganglion). During the operation, all the appropriate procedures were implemented to guarantee the conservation of the nerve branch destined for the tibialis anterior muscle in particular. Once adequate hemostasis was guaranteed, the wound was closed with anatomical layers (the skin with staples). The postoperative course was uneventful, with normal healing and recovery of the surgical wound. Subsequently, a physiotherapy program with kinesitherapy associated with the physical method as electrostimulation was performed.

#### 3.2.3. Follow-Up

At clinical examination, two months after surgery, the patient showed a significant improvement in NRS = 0/10, in MRC of TA = 5/5, EDC and EHL = 4/5, and in peroneal muscle MRC = 5/5, increase in ankle dorsilexion and metatarsophalangeal joint (Figure 2b) and absence of paresthesia in the lateral region of the leg.

## 4. Discussion

Foot drop represents a complex pathological condition, requiring a multidisciplinary approach for appropriate evaluation and treatment [15]. When considering surgical management for the treatment of foot drop, it is first and foremost imperative to establish the cause of the condition. Ganglion cysts responsible for acute-onset foot drop are rare and described in the literature as a single case report [10,14]. The diagnosis of acute foot drop caused by intraneural ganglion cyst of the peroneal nerve is often not immediate and, therefore, the subsequent intervention does not appear to be timely. Over the years, various theories have been developed, which have tried to explain the pathogenesis of this problem [16,17,18]; however, the unifying articular (synovial) theory described by Spinner [19] appears to be the most widely accepted. According to Spinner, endoneural ganglion cysts originate at the level of the nearby joints, such as, for example, the upper tibio-fibular joint, which, through a defect in the joint capsule, creates an extravasation of synovial fluid, which, with a unidirectional mechanism, infiltrates the subepineural space following the articular branch of the peroneal nerve with the appearance of the consequent cystic formation. Various studies support this theory, also demonstrating how a rigorous diagnostic and therapeutic protocol with adequate characterization of the cyst or the presence of an articular peduncle plays a decisive role [20]; our patient, in particular, had significant distension of the superior tibiofibular recess on MRI, supporting Spinner’s theory. The symptoms associated with IG include sensory deficits with paresthesia affecting the lateral surface of the leg and the back of the foot, painful symptoms mainly localized at the head of the fibula, positive Tinel test and motor deficits of the muscles of the anterior region of the leg up to stepping during walking, sometimes characterized by repeated falls [3]. The tendon reflexes, as in the case of our patient, should be normal. The onset is generally gradual, unlike our patient who presented with acute-onset foot drop in the absence of recent traumatic events or previous symptoms. Our case is one of the rare cases present in the literature and, unlike those included in this review [10,13,14], it appears to be the one with the fastest diagnostic and therapeutic classification. Similar to the cases reported by Williams [14] and Stamiris [10], rehabilitative treatment seems to improve the clinical outcomes and, in our case, prompt surgical treatment may have further affected the complete motor recovery. However, even in cases where the symptoms last for months, it may be useful to refer for surgery [21]. A palpable mass lateral to the fibula head may be present [22]. Magnetic resonance imaging and ultrasound examination are thought to be useful tools in diagnostics [10]. On ultrasound examination, an endoneural ganglion cyst appears as a circumscribed hypoechoic formation [23], while on MRI, it appears as a lesion with low signal intensity on T1-weighted images and high signal intensity on T2-weighted images [24]. MRI also shows reproducible signs (tail signs) that can help in the identification of the joint connection, as well as in the differentiation between intraneural and extraneural ganglion cysts [25].

According to Consales et al. [26], a reduction in intraneural pressure has a decisive role in the control of post-operative pain, unlike motor function. The latter is less predictable and must take into account additional factors. Although our patient appeared to have an important functional deficit with an almost complete paralysis, an important motor recovery was observed at the 2-month post-operative follow-up, despite the consistent volume of the cyst and the acute appearance of drop foot. Other authors report partial recoveries after 3 months [13,21], while others report the persistence of slight motor deficits several months after surgery [21,27]. However, there appear to be motor recoveries even seventeen months after surgery [28].

The limitations of this case report may concern the early treatment chosen without guidelines on the acute foot drop. The authors aimed to describe the good clinical outcomes after integrated surgical–rehabilitative treatment, in order to propose a reference diagnostic–therapeutic procedure helping other clinicians. Obviously, further studies on a larger patient population are needed.

## 5. Conclusions

The appearance of acute non-traumatic foot drop secondary to IG remains a rare, dramatic event for patients and requires an appropriate diagnostic–therapeutic framework. IG should be considered in the differential diagnosis of acute foot drop. Ultrasound, MRI, EMG and ENG examinations are valuable instrumental diagnostic elements that contribute to the diagnostic classification. Although we believe that therapeutic timing plays a decisive role in motor recovery, even long-standing cases could benefit from appropriate surgical–rehabilitative treatment.

## Figures and Tables

**Figure 1 jpm-13-01137-f001:**
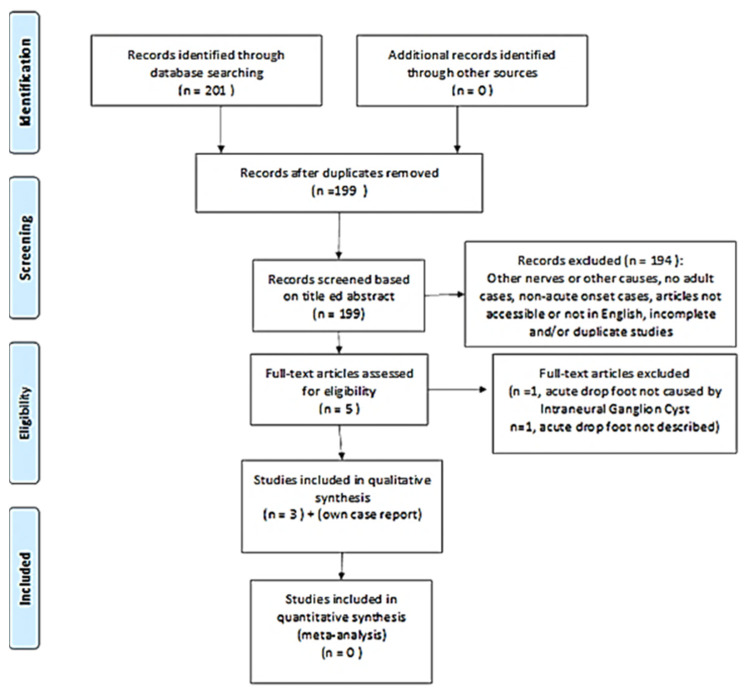
Quality of reporting of meta-analyses (QUORUM) flowchart; the consecutive steps followed during our review of the literature have been illustrated. The reasons for the exclusion of the articles have been mentioned.

**Figure 2 jpm-13-01137-f002:**
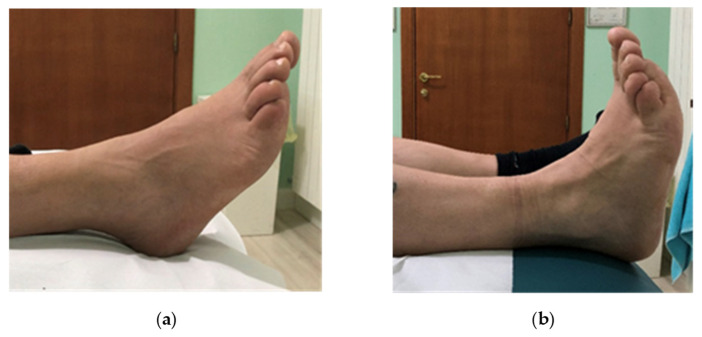
Foot drop. Active ankle dorsiflexion before (**a**,**b**) after surgical and rehabilitation treatment.

**Figure 3 jpm-13-01137-f003:**
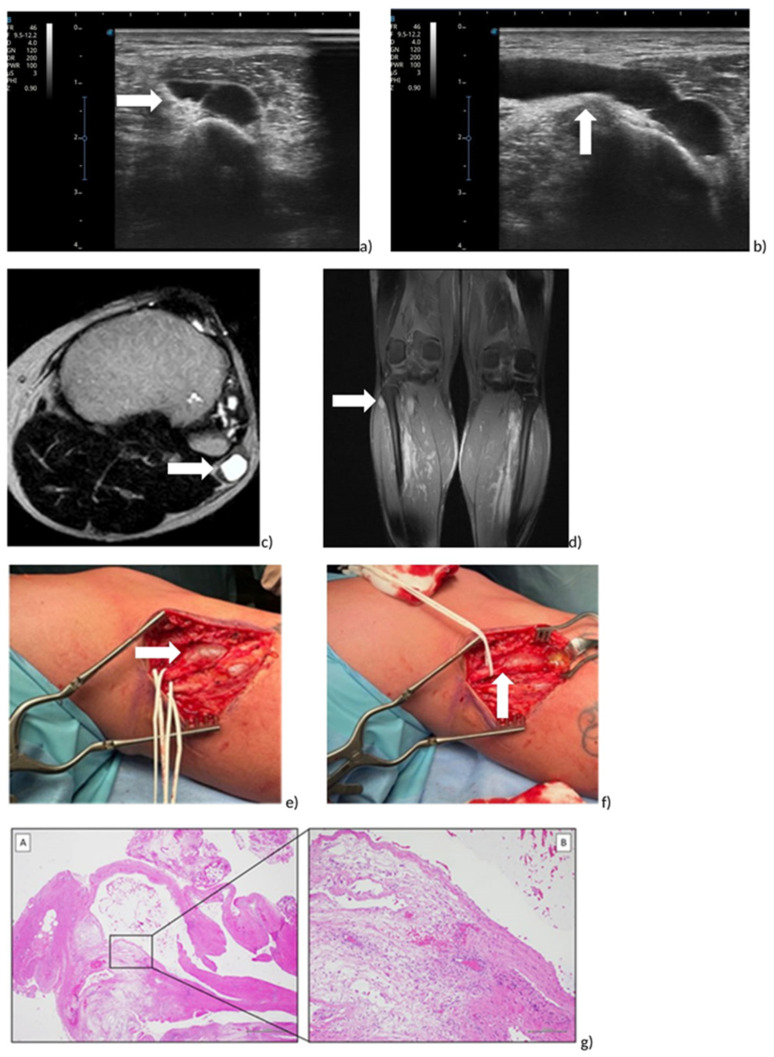
Peroneal intraneural ganglion cyst of right leg. (**a**,**b**) Ultrasound Image on axial (**a**) and sagittal (**b**) plane; (**c**,**d**) RMI on axial (**c**) and coronal (**d**) plane. (**e**,**f**) Intraoperative view. Exposition of common peroneal nerve (**e**) and ganglion cyst (**f**). (**gA**,**gB**) Histological examination: (**gA**) Morphologically, the mass showed a unilocular cystic structure composed of a fibrous wall, with edema and vascular congestion (H&E, 2× magnification, scale bar 1 mm). (**gB**) Higher magnification, in the inset, showed no true “epithelial lining”, absence of cellular atypia, mild inflammatory infiltration and focal hemorrhagic extravasation (H&E, 10× magnification, scale bar 200 µm).

**Table 1 jpm-13-01137-t001:** Review of 3 selected case reports relating to acute foot drop caused by peroneal intraneural ganglion cyst.

First Author (Year)	Study Design, Participant	Pathological Anatomy	Diagnostic Exams	Outcome Measures	Treatment	Follow Up Results
Devon I. Rubin (2004) [13]	Short report, III, 1 participant, age 67, man.	ganglion cyst	MRI, EMG, blood test	complete paralysis of dorsiflexor ankle muscles	surgery	Three months. Ankle dorsiflexion MRC = 3/5, Toe extension MRC = 2/5.
Samantha L. Williams (2022) [14]	Case report, III, 1 participant, age 52, man	not available	MRI, EMG, blood test	Unspecified rating scale. Complete dorsiflexors paralysis of the foot and toes reported	surgery, physiotherapy (with physical therapy and electrostimulation)	One year. Reported complete recovery of strength of the dorsiflexors muscles
Stavros Stamiris (2020) [10]	Case report, III, 1 participant, age 42, man	ganglion cyst	X-ray, EMG, MRI, blood test	TA MRC = 1/5, EHL MRC = 0/5, EDC MRC = 0/5, PM MRC = 2/5.	surgery, physiotherapy	three months. TA MRC = 4/5, EHL MRC = 2/5, EDC MRC = 3/5, PM MRC = 4/5.

Notes: MRC = Medical Research Council; TA = Tibialis Anterior Muscle; EHL = Extensor Hallucis Longus Muscles; EDC = Extensor Digitorum Common Muscle; PM = Peroneal Muscle; MRI = Magnetic Resonance Imaging; EMG = electromyographic examination.

## Data Availability

Not applicable.

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
