# Peer review of "Acute Foot Drop Caused by Intraneural Ganglion Cyst of the Peroneal Nerve: Literature Review and Case Report"

_jpm, 2023, doi:10.3390/jpm13071137_

Round 1

Reviewer 1 Report

I consider it a good manuscript, original and well structured.

Perhaps there may be a small number of bibliographic references, which if increased would improve the quality of the study.

Author Response

Response to Reviewer #1

I consider it a good manuscript, original and well structured. Perhaps there may be a small number of bibliographic references, which if increased would improve the quality of the study.

Reply*: Thanks for the comments. Our manuscript is a case report and this is the main factor limiting the number of references. Moreover, it is associated with literature review on acute onset of foot drop, which is a less studied and frequent topic.  These two elements negatively affect the presence of articles and the possibility of citations. However, we added 3 articles ([13], [14], [15]) which were cited in the review and not previously included in the references

Reviewer 2 Report

Dear authors I read the manuscript with interest. Please consider the following for an updated version

1. Did you perform EMG examination after the initial 10 days of management? If yes please let us know the result

2. Postoperative how did you document the peroneal nerve status?

3. Intraneural ganglion cyst often result to progressive neurological deficits. Please provide an explanation for the acute onset of the drop foot in your case

Minor editings should be considered

Author Response

Response to Reviewer #2

Dear authors I read the manuscript with interest. Please consider the following for an updated version

  1. Did you perform EMG examination after the initial 10 days of management? If yes please let us know the result

Reply*: Thanks for the query. The patient was undergone to EMG examination after three days from our evaluation (about 13 days from the clinical onset) and we described the results in the text at line 135-137.

After surgical treatment, the patient didn’t perform EMG exam, because patient’s follow up was based on clinical outcomes.

  1. Postoperative how did you document the peroneal nerve status?

R*: Significant clinical improvement guided patient’s follow up. Moreover, patient also performed an ultrasound examination, less expensive than other ones in terms of cost and time.

  1. Intraneural ganglion cyst often result to progressive neurological deficits. Please provide an explanation for the acute onset of the drop foot in your case

R*: We try to give an explanation on the text in Discussion at line 180-192. Probably the acute clinical onset does not correspond to an acute anatomical-pathological pattern but progressive up to a functional tolerance of the nerve.

Comments on the Quality of English Language

Minor editings should be considered

Reviewer 3 Report

The paper deals with a interesting topic: acute foot drop caused by intraneural ganglion cyst of the peroneal nerve. As noted by the authors "the onset of the foot drop can be secondary to an acute traumatic event that induces the onset of paralysis, but it may be also a primary symptom with an acute onset and in the absence of a previous traumatic event in the patient apparently healthy. Therefore, a careful diagnostic therapeutic procedure must be performed for an adequate approach and treatment is necessary".

Thank you for the opportunity to review this study. However, I have some suggestions for improving the manuscript:

1. There is no information in the manuscript whether the patient used any form of rehabilitation before the surgery;

2. There is no information whether the patient practiced any sports - maybe there was some injury that caused the ganglion to develop;

3. In line 164, the term/word "physio-kinesitherapy" is used, it is a wrong term - please correct it for physiotherapy and specify that it is about activities involving exercises with patients, i.e. "kinesiotherapy" and physical methods (electrostimulation);

4. The subsection "limitations of the study" is missing from the manuscript;

5. Subsections 2.2 and 3.1 contain the same information - please change the paragraphs or delete them to remove redundant repetitions;

6. Please remove the double brackets on line 91 - "The following search string was used: ((peroneal ...";

7. In their manuscript, the authors distinguished 3 articles in which the researchers using appropriate treatment achieved beneficial effects in this group of patients. In two of them surgery and physiotherapy were used, and in one only surgery was used. The authors' study also shows that surgery in conjunction with physiotherapy brings beneficial effects in patients with acute foot drop caused by intraneural ganglion cyst of the peroneal nerve. Therefore, in the Conclusions section, please refer to this information.

8. In the discussion section, it would be useful to refer to more literature and compare with similar studies. After all, the research does not have to be identical, but it is worth emphasizing that this is a problem for which various actions have been taken, with different results.

9. In Table 1, references from references should be added to references.

10. In line 63, is the term "PRISMA 12" correct? - should it be "PRISMA 2020"?

11. On line 69, please standardize and change the capital letters "Foot Drop" to "foot drop";

12. In line 91, instead of "Diagram 1" should be Figure 1. (other terms are misleading) - please correct;

13. Please explain the abbreviation "PICO", not all readers know what it stands for;

14. Please make a space between Scale and [NRS] on line 118: "(Numerical Rating Scale[NRS] 8/10)";

15. On line 119, there is no need to re-write the full phrase: (Medical Research Council scale: MRC=1/5) - please enter only the MRC abbreviation as it has been expanded before;

16. Line 123 (figure 2a) - earlier "Figure" was capitalized, similarly in lines 156 and 169 - please harmonize;

17. On line 139, please change from "and after b) surgical and rehabilitation treatment" to "and b) after surgical and rehabilitation treatment".

18. Throughout the manuscript, and above all in the discussion, lines 176, 180 - please correct the footnotes by placing them in one parenthesis and separating them with a comma, i.e. [10, 14], not [10][14];

19. On line 200, "Magnetic resonance imaging (MRI)", please leave the expansion or abbreviation as the abbreviation has already been explained;

20. In subchapter 2.2.3. (line 168), ECD and ELA abbreviations are not explained - please correct.

21. Information about the patient and applied therapy from subchapter 3.2.1. should be transferred to the Material and Methods chapter, in the Results section the treatment results should be described.

Author Response

Response to Reviewer 3

The paper deals with a interesting topic: acute foot drop caused by intraneural ganglion cyst of the peroneal nerve. As noted by the authors "the onset of the foot drop can be secondary to an acute traumatic event that induces the onset of paralysis, but it may be also a primary symptom with an acute onset and in the absence of a previous traumatic event in the patient apparently healthy. Therefore, a careful diagnostic therapeutic procedure must be performed for an adequate approach and treatment is necessary".

Thank you for the opportunity to review this study. However, I have some suggestions for improving the manuscript:

  1. There is no information in the manuscript whether the patient used any form of rehabilitation before the surgery;

Reply*: Thanks for the query. Before the surgery, the patient performed rehabilitation treatment including kinesiotherapy associated to electrostimulation. We added this information in the text at line 153.

  1. There is no information whether the patient practiced any sports - maybe there was some injury that caused the ganglion to develop;

R*: thanks for the query. The patient didn’t practice any sport.

  1. In line 164, the term/word "physio-kinesitherapy" is used, it is a wrong term - please correct it for physiotherapy and specify that it is about activities involving exercises with patients, i.e. "kinesiotherapy" and physical methods (electrostimulation);

R*: Thanks for the correct observation. We revised the text at line 166.

  1. The subsection "limitations of the study" is missing from the manuscript;

R*: thanks for the comment and suggestion. As case report, the limitations didn’t concern the study design but, probably, the type of the treatment chosen without guidelines on the acute foot drop. We hope that this case report, together with those few others, can propose a reference diagnostic-therapeutic procedure for colleagues. Obviously future studies on larger number of cases are needed. We added this subsection at line 222-226.

  1. Subsections 2.2 and 3.1 contain the same information - please change the paragraphs or delete them to remove redundant repetitions;

R*: Thanks for the comment and suggestion. We deleted the repetitions and revised the text.

  1. Please remove the double brackets on line 91 - "The following search string was used: ((peroneal ...";

R*: Thanks for the correction. We revised the text.

  1. In their manuscript, the authors distinguished 3 articles in which the researchers using appropriate treatment achieved beneficial effects in this group of patients. In two of them surgery and physiotherapy were used, and in one only surgery was used. The authors' study also shows that surgery in conjunction with physiotherapy brings beneficial effects in patients with acute foot drop caused by intraneural ganglion cyst of the peroneal nerve. Therefore, in the Conclusions section, please refer to this information.

R*: Thanks for the suggestion. We underlined the importance of integrated surgical-rehabilitative treatment in discussion (line 224) and conclusions (234).

  1. In the discussion section, it would be useful to refer to more literature and compare with similar studies. After all, the research does not have to be identical, but it is worth emphasizing that this is a problem for which various actions have been taken, with different results.

R*: Thanks for the suggestion. We have expanded the text in the discussion at line 201-203

  1. In Table 1, references from references should be added to references.

R*: Thanks for the comment. We revised the text adding the references.

  1. In line 63, is the term "PRISMA 12" correct? - should it be "PRISMA 2020"?

R*: Thanks for the correct observation. We revised the text.

  1. On line 69, please standardize and change the capital letters "Foot Drop" to "foot drop";

R*: Thanks for the comment. We revised the text.

  1. In line 91, instead of "Diagram 1" should be Figure 1. (other terms are misleading) - please correct;

R*: Thanks for the suggestion. We revised the text.

  1. Please explain the abbreviation "PICO", not all readers know what it stands for;

R*: Thanks for the comment. We integrated this information in the text.

  1. Please make a space between Scale and [NRS] on line 118: "(Numerical Rating Scale[NRS] 8/10)";

R*: We revised the text.

  1. On line 119, there is no need to re-write the full phrase: (Medical Research Council scale: MRC=1/5) - please enter only the MRC abbreviation as it has been expanded before;

R*: We revised the text.

  1. Line 123 (figure 2a) - earlier "Figure" was capitalized, similarly in lines 156 and 169 - please harmonize;

R*: We revised the text.

  1. On line 139, please change from "and after b) surgical and rehabilitation treatment" to "and b) after surgical and rehabilitation treatment".

R*: We revised the text.

  1. Throughout the manuscript, and above all in the discussion, lines 176, 180 - please correct the footnotes by placing them in one parenthesis and separating them with a comma, i.e. [10, 14], not [10][14];

R*: Thanks for the suggestion. We revised the text.

  1. On line 200, "Magnetic resonance imaging (MRI)", please leave the expansion or abbreviation as the abbreviation has already been explained;

R*: We revised the text.

  1. In subchapter 2.2.3. (line 168), ECD and ELA abbreviations are not explained - please correct.

R*: Thanks for the correction. We revised the text correcting ECD in EDC and ELA in EHL (previously explained)

  1. Information about the patient and applied therapy from subchapter 3.2.1. should be transferred to the Material and Methods chapter, in the Results section the treatment results should be described.

R*: Thanks for the suggestion. We understand reviewer’s correct indication but we would like to leave this setting in the text to make the description of the case report easier to read and more cohesive and understandable.

Round 2

Reviewer 3 Report

I thank the authors for responding to my comments/suggestions.

Please, correct the position of the title of table 1 - it should be above the table, not below.

Author Response

I thank the authors for responding to my comments/suggestions.

Please, correct the position of the title of table 1 - it should be above the table, not below.

R*: We revised the text according to the reviewer's comment, placing the title of table 1 above the table.